# Content Validity of a Scale Designed to Measure the Access of Older Adults to Outpatient Health Services

**DOI:** 10.3390/ijerph191610102

**Published:** 2022-08-16

**Authors:** Gerardo Santoyo-Sánchez, César Merino-Soto, Sergio Flores-Hernández, Blanca Estela Pelcastre-Villafuerte, Hortensia Reyes-Morales

**Affiliations:** 1School of Public Health of Mexico, National Institute of Public Health, Avenida Universidad 655, Santa María Ahuacatitlán, Cuernavaca 62100, Morelos, Mexico; 2Psychology Research Institute, San Martin de Porres University, Avenue Tomás Marsano 232, Lima 34, Peru; 3Center for Evaluation and Surveys, National Institute of Public Health, Avenida Universidad 655, Santa María Ahuacatitlán, Cuernavaca 62100, Morelos, Mexico; 4Center for Health Systems Research, National Institute of Public Health, Avenida Universidad 655, Santa María Ahuacatitlán, Cuernavaca 62100, Morelos, Mexico

**Keywords:** access to care, older adults, outpatient health services, primary health care, validation study

## Abstract

The objective of this work was to validate the content of a scale formulated in Spanish for older adults in Mexico, with the aim of comprehensively measuring the access of this population group to outpatient primary-care services. To this end, we carried out a methodological content-validity study in four stages: (1) construction of the scale; (2) evaluation of item legibility; (3) quantitative content evaluation by two groups of judges selected by convenience: participant-judges including older adults with adequate reading comprehension, surveyed in person (*n* = 23), and expert-judges comprised of researchers specialized in the fields of health services, psychometrics and aging, surveyed online (*n* = 7); and (4) collection of qualitative feedback from several of the participant-judges (older adults, *n* = 4). The content was validated both by sequentially examining the level of consensus in the responses of both groups of judges, using the Tastle and Wierman method, and by calculating Aiken’s Validity Coefficient with a 90% confidence interval. The scale contained 65 items pertaining to 10 dimensions of two major constructs: accessibility (*n* = 39) and personal abilities (*n* = 26). Five items were eliminated in accordance with the minimum-consensus criterion (0.5). This is the first psychometric scale to be developed in Mexico with the view of integrating the characteristics of health-care services and the abilities of the older adults in a single questionnaire designed to measure the access of this population group to outpatient primary-care services.

## 1. Introduction

Global population aging, reflected in a growing number of older adults worldwide, is imposing new demands on health systems [1,2]. The number of older adults in Mexico, defined as individuals aged 60 and above [3], has exceeded 15 million (equivalent to 12% of the population), and is expected to continue rising at a rate three times as high as that of the general population [4,5]. Meeting the health needs of this rapidly swelling population group requires effective access to services and the provision of equitable care [6]. Utilizing outpatient primary-care services (OPPCSs) has proved to be the most effective strategy for meeting the multiple health needs of older adults [7]. In offering comprehensive care for the prevention and resolution of most health problems [8,9], these services are key to maintaining or improving the health and longevity of older adults.

It has been shown that lack of access to health services is related to the limited availability and distant geographic locations of primary-care facilities [10,11,12]. With regard to older adults specifically, low levels of access to OPPCSs have been reported among those without Social Security coverage [13,14]; they have also been associated with the lack of public transportation, as well as with the physical and financial dependence of the elderly on others [15]. Finally, a growing body of evidence indicates that cultural and geographic barriers limit access of older adults to services of this type [16,17,18]. However, despite numerous efforts to understand the barriers faced by this population group in obtaining care, research has failed to integrate the complex factors affecting older adults’ access to OPPCSs into a single report; studies have yet to clarify whether difficulties in accessing services are more a result of limitations in personal abilities (e.g., perceiving, seeking, reaching, paying, and engaging), the characteristics of health-care systems, organizations, and providers, or the interaction between the two.

Conducting a comprehensive measurement of older adults’ access to OPPCSs poses at least three major challenges. First, it is difficult to define and quantify access [19,20]. No one concept of access has yet been widely accepted; rather, different perspectives have been adopted, with only a few actually addressing the numerous complexities involved in accessing services. For example, Salkever’s model emphasizes service characteristics, while that developed by Bashshur et al. regards personal abilities solely as confounding factors [21,22]. Second, the indicators currently used for measuring access are limited in that they merely explore whether or not respondents are familiar with, have ready, or use services [23,24,25]. They do not measure whether the characteristics of services, providers, and systems interact with people’s abilities or with the characteristics of households and communities, as has been proposed in more inclusive conceptual models [26]. Finally, a third limitation concerns the fact that people tend to underestimate the importance of access measurements based on the perceptions of older adults [27]. In this respect, however, evidence has shown that perceptions regarding access provide greater information about the phenomenon than measures of actual access do [27,28].

Few internationally available psychometric scales measure access in an integrated manner and considering patient perceptions [29]. Our literature review revealed that no scale had been specifically designed to measure the perceived access of older adults to OPPCSs comprehensively. In constructing such a scale, we found that using a structured theoretical model such as Standards (*Standards for Educational and Psychological Testing*) facilitated decision making while planning and assessing the interaction among the validity evidence variable [30,31]. During validation processes, the first kind of evidence normally analysed is content validity, defined as the degree to which the contents of a test are congruent with its purposes [32]. Assessments of content validity are largely a matter of judgement and often involve the following activities: (1) a priori efforts on the part of those developing the scale, e.g., conceptualization based on the elaboration or selection of elements from existing literature and theories; and (2) a posteriori efforts on the part of a panel of judges recruited to evaluate the proposed texts and determine their relevance in relation to the content domain [31,33].

In light of the foregoing, we undertook this study to generate content-validity evidence as a basis for creating an innovative scale aimed at comprehensively measuring the access of older adults to OPPCSs in Mexico. We accomplished this by (a) identifying and conceptualizing relevant constructs; (b) searching for and selecting available measures for the constructs; (c) determining the operational process most suitable for the construction of items; (d) collecting the opinions of the participant- and expert-judges on the relevance and clarity of the items; and (e) obtaining qualitative feedback from the participant-judges. We also performed a quantitative evaluation of item legibility.

## 2. Materials and Methods

We used a methodological content-validity study design within the Standards framework [30,31]. As shown in Figure 1, we carried out the study in four stages related to six content-validity components: (1) construction of the scale; (2) evaluation of item legibility; (3) quantitative content evaluation by two types of judges (participant-judges and expert-judges); and (4) gathering of qualitative feedback from the participant-judges.

### 2.1. Stage 1. Construction of the SCALE

Construction specifications required that the scale (1) be generic, in other words, that it contain no specific references to the respondents’ surroundings, illnesses, or OPPCSs; (2) provide a cross-sectional measurement of access, excluding changes and effects over time; (3) has the potential to measure access broadly by encompassing the dimensions and elements of both supply and demand; (4) allows for rating the items according to an unweighted Likert-type scoring system; (5) ensures the integration/interaction of its constructs and dimensions; (6) orders the items according to the dimensions they represent; and (7) allows for supervised administration.

#### 2.1.1. Identification and Conceptualization of Constructs

We determined the constructs and dimensions of the scale a priori [34] according to Levesque’s access model [26]. This approach conceptualizes access based on two major constructs, each containing five dimensions: (1) accessibility—approachability, acceptability, availability and accommodation, affordability, and appropriateness; and (2) personal abilities—the capacity to perceive, seek, reach, pay for services, and engage with care providers. Based on these references and the theoretical information on constructs drawn from the literature [35,36,37], we conceptualized accessibility as the degree to which older adults can easily reach (approachability), accept (acceptability), and avail themselves (availability and accommodation) of outpatient services that are affordable (affordability) and suitable (appropriateness) for their health needs [26,36,37]. We assumed that an interaction existed between the perceptions of older adults concerning the complexity/difficulty inherent in these actions and those concerning their own capacity to execute them. We defined the personal abilities construct as the capacity of older adults to perceive the need for health (ability to perceive), seek adequate care options (ability to seek), reach outpatient services (ability to reach), cover their costs (ability to pay), and engage with care providers in adhering to the treatment (ability to engage) [26,35]. In this regard, we presupposed the existence of variability in individual capacities and strategies for seeking services.

#### 2.1.2. Search, Selection, and Adaptation of Construct Measurements

Using the deductive method, that is, inferring from available materials which content would be suitable for the scale [34], we searched for items across three main sources, under the assumption that they possessed adequate metric properties: (1) national surveys on health and aging conducted in Mexico during 2018; these surveys have been a source of population information representative at national/regional levels and with demonstrated comparability over time in the last two decades in Mexico [38,39]; (2) standardized measurements of accessibility and personal abilities [36,40,41]; and (3) empirical studies on access to health services [35,42,43,44,45,46,47,48,49]. Based on these measures, we then classified (grouped) and selected the items consistent with our operational definitions. Finally, the principal researcher eliminated redundant items and adjusted the drafting of those remaining, preserving simple spoken language and short sentences [50] expressed affirmatively [51] and without qualifiers or ambiguous terms [34,52,53].

#### 2.1.3. The Operational Process Used to Construct New Items

Our method featured a specific process for making decisions and reaching consensus among research team members (all the authors). Team members possessed experience in the fields of access to health services and psychometrics. We held regular meetings over a period of four months in order to convert the qualitative components of the model a priori into items. The questions were drafted according to the following grammatical criteria: formulating affirmative sentences [51] of approximately the same length and using simple spoken language [50] without qualifiers or ambiguous terms [34,52,53]. After drafting the items, the research team discussed each one separately, edited those requiring revision, and ensured that all accurately reflected the qualitative themes of the model. The team was particularly attentive to constructing items that would capture the experiences of older adults as they sought and accessed health services [34,36].

### 2.2. Stage 2. Evaluation of the Linguistic Legibility of the Items

One of our strategies for validating content clarity consisted of evaluating item legibility. For this purpose, we used three formulas featuring different approaches: (1) the Fernandez-Huerta Legibility Formula (206.84 − 0.60P − 1.02F), (2) the Szigriszt-Pazos Perspicuity Index (207 − (62.3 s): p − (p:f)), and (3) the INFLESZ Scale, or Flesch-Szigriszt Index (206.835 − (62.3S/P) − (P/F)). The first, created for Spanish speakers, excludes typographical elements that can influence reading ease [54]. The second, designed for English speakers, was adapted to Spanish employing the Flesch formula [55]. Finally, the INFLESZ Scale, adapted from the Szigriszt-Pazos Index, takes into account sample size and text variability [56]. We carried out the analysis in two phases. First, we submitted each item, without punctuation marks, to each of the three formulas. Second, we subjected the group of items in each construct (accessibility and personal abilities) to the three formulas, first separately and then collectively, in order to obtain a total legibility estimate. For all calculations, we used the Legible.es parser, a free Python script publicly available on the Internet under General Published License 3 [57].

### 2.3. Stage 3. Quantitative Content Evaluation

We conducted a quantitative evaluation of the items in the preliminary version of the scale as developed during the first two stages of our content validation process; to this end, we used two indicators: (1) clarity, defined as the degree to which individuals can read and understand a text, assuming a consistent level of reading comprehension among respondents; and (2) relevance, defined as the degree to which the items represent the dimensions of interest. The first was evaluated by both groups of judges, and the second by the expert-judges alone.

#### 2.3.1. Participants

Selected by convenience (January 2021), participant-judges exhibited similar characteristics to those of the older adults for whom the scale was designed. Their number was determined according to common practice in content validation exercises (a minimum of five individuals is normally recommended) [34]. Inclusion criteria consisted of older adults (1) aged 60 and above, (2) lacking Social Security coverage, and (3) residing in urban areas (>2500 inhabitants) [58], in Mexico. Excluded from analysis were older adults with mental disability or functional dependence that prevented them from participating, as well as those who refused to provide verbal informed consent and/or whose caregivers or accompanying family members refused to be interviewed. Of the 53 older adults who agreed to participate, seven were eliminated because they misunderstood the instructions and/or failed to complete the survey; the 46 who completed the survey were then subjected to a further inclusion filter regarding adequate reading comprehension (see Appendix A). The final study sample of participant-judges included 23 older adults with adequate reading comprehension (see Section 3).

The panel of expert-judges, also selected by convenience (January 2021). Selection criteria were: proficient in Spanish; and expertise in the fields of health services, scale development methods, and aging. Their number was determined in the same manner as that of the participant-judges. From a panel of 13 specialized professors–researchers from universities and public health institutions in Mexico and the United States, only nine were able to participate individually; of these, two were eliminated for non-compliance with the inclusion criteria. A total of seven expert-judges remained in the study.

#### 2.3.2. Instruments

We administered two instruments to the participant-judges:

(1) A background characteristics/literacy proxy questionnaire served to collect self-reported data on two types of information: (a) the socioeconomic and demographic traits of the judges (age, sex, marital status, level of education, occupation, health needs, and care seeking), and (b) their levels of literacy. The literacy proxy included a four-item Self-Assessed Literacy Index (SALI) [59] and a one-item reading comprehension test (the Single-Item Literacy Screener, SILS) [60]. The SALI, our standard criterion, assessed reading, understanding, speaking, and writing skills in Spanish. Scores for the four SALI responses were added up, while the one response to the SILS (“How often do you have difficulty understanding written information or instructions?”) was rated on response categories from one (“Always”) to five (“Never”).

(2) Clarity form. Using response categories from one (“Not at all understandable”) to five (“Fully understandable”), this document measured the clarity levels of the 65 items in the preliminary version of the scale. The items were ordered according to the 10 dimensions of the access constructs they represented. We included a simple definition of each dimension to ensure that the participant-judges understood what the items referred to. Additionally, at the end of the form, we provided a section for respondents to write qualitative suggestions on the clarity of the items.

For the expert-judges, we used the following instruments:

(1) A Background Characteristics questionnaire served to collect their sociodemographic and academic information (sex, country of residence, institutional affiliation, and number of publications in peer-reviewed journals).

(2) A Clarity and Relevance form allowed for measuring these two indicators throughout the 65 items in the preliminary version of the scale. The items were grouped under the 10 dimensions of the access constructs they represented. Responses were rated on categories from one (“Not at all”) to seven (“Completely”). To ensure that the expert-judges understood the meanings of the dimensions analysed, we included their operational definitions as well as a reference to the access model on which the study was framed. At the end of the form, we added a section for respondents to provide qualitative suggestions on the elements measured.

#### 2.3.3. Procedures

We collected information on participant-judges using a face-to-face survey administered by previously trained field personnel under supervision by a researcher (first author G. Santoyo, GS). In order to identify potential participants, we carried out sampling in the urban area where the study was conducted, and randomly visited all dwellings until the required number of people meeting the inclusion criteria had agreed to participate. Each person was invited to join the study after providing informed consent, including that pertaining to health security in the face of COVID-19; participation was voluntary and non-remunerative. We followed the same administrative procedures and order of presentation of materials for all participant-judges: we obtained informed verbal consent, conducted an assessment of background characteristics, and provided a literacy proxy and clarity form (supervised and self-administered, with paper and pen/pencil).

The team contacted the expert-judges through their private and/or institutional e-mail addresses. We sent each prospective judge a letter with information about the study, a description of the content validation process, and an invitation to participate on the understanding that there would be no compensation. Once they had agreed to take part, they received the Background Characteristics questionnaire and the Clarity and Relevance form. Potential participants were given four weeks to complete the survey; two reminders (scheduled one week apart) were sent to those who did not respond.

### 2.4. Stage 4. Qualitative Feedback on Content

Following the quantitative evaluation of the content of the scale, we conducted interviews with participant-judges (May 2021) selected by convenience from among the subsample of those with adequate reading skills (see Section 3). To conduct the interviews, a research assistant visited the homes of the selected participant-judges and provided them with a mobile telephone to connect them with a trained researcher. The assistant also gave each participant-judge a hard copy of the initial draft of the scale. During the interviews, the participant-judges provided feedback concerning the relevance of the items for each dimension of the accessibility and personal abilities constructs. They were also asked if they found the various items clear and/or if they thought any changes to the wording were needed. The research group used a pragmatic approach to the analysis regarding any required modifications. The team in charge recorded all interviews after obtaining informed verbal consent, including that which pertaining to health security in the face of COVID-19.

### 2.5. Data Analysis

To analyse the relationship between the results of the two methods used to determine reading comprehension, we utilized Spearman’s correlation (Stage 3) (using SPSS^®^/PC Ver. 26) [61] and performed a quantitative content analysis in two phases: within each category of judge (*intra-judges*) and comparing the two categories (*inter-judges*) (Stage 3). In the first phase, the team examined the levels of consensus of the two groups—expert-judges and participant-judges—based on Tastle and Wierman’s consensus indicator [62] (using R Studio Ver. 4.0.3) [63]. The evaluation of the differences in response trends was of great practical significance, as it avoided the loss of statistical power in all subsequent inferential evaluation tests [64]. Only items with a consensus ≥ 0.50 were estimated using Aiken’s V coefficient to assess the two indicators of content validity (clarity and relevance), with asymmetric confidence intervals (90% CIs) [65,66]; items with a CI limit < 0.60 [67] were eliminated. The team subsequently validated the content for the five dimensions of each construct (accessibility and personal abilities) using the method outlined by Merino-Soto and Livia [68]. In the second phase, the study compared the perceptions of the expert- and participant-judges regarding the clarity indicator of content validity, using the confidence interval method for determining differences between coefficients known as Aiken’s V Coefficients [69]; all analyses were performed using the SPSS^®^/PC Ver. 26 statistical package [61].

## 3. Results

### 3.1. Stage 1. Construction of the Scale

We selected 43 out of the 65 initial items from the three sources referred to in Substage 1.2 of the method and constructed 22 according to the process described in Substage 1.3. As this study adapted the wording of the 43 items selected, none were identical to those found in the three sources. All selected/adapted and constructed items were grouped into two components according to the conceptualization of the constructs described in Substage 1.1 of the method: accessibility (39 items) and personal abilities (26 items). The accessibility construct was divided into five dimensions: approachability (8 items), acceptability (5 items), availability and accommodation (13 items), affordability (6 items), and appropriateness (7 items). The personal abilities construct was also divided into five dimensions: the ability to perceive (6 items), seek (4 items), reach (5 items), pay (6 items), and engage (5 items). The items were arranged according to the set of elements provided by Levesque’s model (see Table 1 and Table 2).

### 3.2. Stage 2. Evaluation of the Linguistic Legibility of the Items

To analyse the items—those grouped by dimension as well as those appearing in a complete text—we categorized them as “very easy”, “easy”, “fairly easy”, and “somewhat easy” according to the linguistic legibility scores, as defined by Fernández-Huerta, Szigriszt-Pazos, and INFLESZ. These ratings revealed that the 65 items of both constructs contained no grammatically complex structures and were readable at the fourth-to-sixth-grade levels. Extrapolating these findings in a Mexican context, we concluded that the items were legible for older adults with either complete or incomplete elementary-level (basic) education [70].

### 3.3. Stage 3. Quantitative Evaluation of the Content of the Scale

#### 3.3.1. Participants

During the process of identifying participant-judges with adequate reading comprehension, we found that SILS and SALI (our standard criterion) were moderately related with this ability (*rho* = 0.367, *p* = 0.012). However, when correlating SILS exclusively with the SALI item that directly measured reading comprehension, we obtained a stronger relationship (*rho* = 0.449, *p* = 0.01), demonstrating that SILS was a suitable test for measuring reading comprehension. In administering SILS to the 46 participant-judges who had completed the survey (with a duration of ~45 min each), we found that only 23 (50%) achieved adequate levels of reading comprehension, leaving a final sample of 23 individuals for analysis. A higher percentage of women (*n* = 13; 56.52%) tested at an adequate level, and the overall average age for the entire sample was 67 years (percentiles 25–75, 64–72). In examining the range of dichotomized reading comprehension levels (M_R_) and the clarity scores obtained for all 65 items in the scale, we found that the judges who were classified as enjoying adequate reading comprehension provided higher clarity ratings than those with limited comprehension. To assess the sizes of the differences in levels of reading comprehension between the judges with adequate vs. limited ability, we estimated the probability coefficient of superiority: *r_MW_* = Z/√*n*. In using the following interpretative standards: no effect *(r_MW_* ≤ 0.0), small effect *(r_MW_* ≥ 0.10 to < 0.30), medium/moderate effect *(r_MW_ ≥* 0.30 to <0.50), and large/substantial effect *(r_MW_* ≥ 0.50) [71], we observed predominantly small-to-medium effects ranging from −0.126 for item 10 to −0.468 for item 38.

As for the expert-judges (*n* = 7, eliminated cases: 2), the final sample was comprised mainly of women (*n* = 6; 85.71%), with an average of 22 articles published in peer-reviewed journals (percentiles 25–75, 20–70).

#### 3.3.2. Intra-Judge Estimates

*Estimation of consensus*. The participant-judges, who evaluated only clarity, reported values ranging from 0.488 (weak consensus) to 0.816 (strong consensus). Only two items (It10 and It11) within the “acceptability” dimension of the accessibility construct demonstrated dispersed responses, with a consensus < 0.5. Meanwhile, among the expert-judges, three items (It4, It49, and It54) obtained dispersed responses for clarity vs. five items (It10, It45, It49, It54, and It60) for relevance, reaching a consensus value < 0.5. Overall, participant- and expert-judges expressed concentrated, unimodal, and similar opinions. With a consensus value < 0.5 for relevance, five items (It10, It45, It49, It54, and It60) proved unacceptable, reducing the total number of items in the scale to 60.

*Estimation of content validity*. Once the levels of consensus were calculated, we assessed clarity and relevance for the 60 remaining items (results available in Figure 2 and Figure 3, and Table 3). The coefficients for all 60 items were statistically significant with respect to the Aiken minimum criterion (0.60). For both clarity and relevance, the closest coefficients occurred primarily in the “approachability” dimension of the accessibility construct. The lowest interval did not meet the minimum Aiken criterion for four clarity items and one relevance item; these were considered minor discrepancies, however, calling only for revisions in wording (limits are underlined in Table 3 and highlighted in Figure 2).

#### 3.3.3. Inter-Judge Estimates

In using Aiken V coefficients to estimate the differences in content-validity scores between the participant- and expert-judges, we obtained confidence intervals of zero employing the Merino method [69], which indicated no statistically significant discrepancies between the two groups of judges. However, we noted slight discrepancies for various items pertaining to the accessibility construct, and, to a lesser extent, to the personal abilities construct, with these items being perceived as less clear by participant-judges.

### 3.4. Stage 4. Qualitative Feedback on Content

We conducted telephone interviews with four participant-judges (three women and one man) who provided feedback, for 30 min each, on the relevance and clarity of the preliminary version of the scale. As regards relevance, interviewees generally showed a positive attitude towards the tool, and found the constructs and their dimensions relevant for measuring the access of older adults to OPPCSs. On the negative side, some respondents found the number of items excessive, while three also found the “approachability” dimension of the accessibility construct not particularly relevant. Regarding the clarity of items, most respondents preferred the terms “health services, health facilities, and clinics” to “outpatient primary-care services”; they recommended improving the clarity in the wording of a number of items.

## 4. Discussion

The scale developed and validated in terms of its content in this study is the first in Mexico to address the concept of access comprehensively, placing special emphasis on the perceptions of older adults. In accordance with the framework utilized [26], we analysed access on the basis of ten broad dimensions of these constructs pertaining to both supply (accessibility) and demand (personal abilities). While separately influenced, the two interact, giving rise to a sequence of transitional steps that culminate in access to health care. Focused on the individual, the model places at the centre of analysis the path patients must navigate to gain access to the care they need.

In the course of developing the scale, we identified other standardized measures within the same framework on access to health care [36,40,41]. However, these estimated only portions of the model constructs, and were not specifically focused on older adults. Furthermore, the measurements were carried out within limited contexts among well-defined populations (e.g., in Canada), featuring psychometric properties that must be evaluated for their applicability to other populations. For these reasons, after reviewing the literature, we confirmed that a complete content evaluation focused on Mexican older adults was clearly necessary, and thus undertook the present study in pursuit of this objective.

The content validity of the scale was enhanced through the implementation of several content development and validation processes. For instance, the framework of the Standards theoretical validity test, considered as representing the best practice in the field of psychometrics, facilitated the planning and systematization of analyses for seeking evidence of validity [30,31]. Within this framework, all proofs of validity contributed directly to content validity, perceived in terms of degrees. In adopting this framework, we abandoned the traditional conceptualization that considers validity to consist of various types (e.g., construction, criterion, and apparent) and/or to be restricted to a dichotomous interpretation (whether or not it exists) [31].

The initial stages of our work included an a priori [34] process aimed at identifying the constructs (accessibility and personal abilities) and dimensions of the scale. Efforts kicked off with a literature review as a means of ensuring adequate conceptual definitions [34,72]. It is worth mentioning that, unlike the definition of first-contact accessibility adopted by other studies using the same model, our concept of accessibility was not limited to medical care [26,36,37]. Primary care must include multidisciplinary teams which, in conjunction with physicians, offer comprehensive services according to the needs of the individual [73]. Specifically with regard to population aging, it has been shown that first contact increasingly involves an array of providers who must be able to respond to a number of health needs, not necessarily pathological [74].

After selecting/adapting and generating the items for the scale, we evaluated their legibility as an additional methodological component of assessing content-clarity validity. This was carried out prior to the clarity evaluation performed by the participant- and expert-judges. Legibility, evaluated in terms of how easily individuals can read and understand a text [75], can seriously affect the validity and reliability of a scale. It has been reported that items that are difficult to read are largely responsible for the failure of older adults with limited education and poor reading skills [76,77] to respond. Therefore, although this additional measure may be considered a proxy for the legibility of the items (as a result of the limited length of the text analysed), its use reinforced content-clarity validity by facilitating the elimination of grammatically complex structures that could have generated confusion or have led to a lack of response from the population of interest (older adults).

A strength of this study lies in the participation of judges with characteristics similar to those for whom the scale was developed (older adults). Previous studies have used expert-judges more often than members of the population of interest to evaluate content validity [34]. However, we decided to evaluate item clarity using both types of judges (experts and participants) simultaneously. The engagement of the population of interest as evaluators has become an important source of content validation that positively impacts the results of analyses [34,64]. For example, measurements of health outcomes require triangulating the perceptions of participant- and expert-judges to achieve a high level of adequacy regarding content relevance, importance, and clarity. Thus, a sole source of content evaluation may be insufficient, even more so when discrepancies between the two types of judges are detected [64]. In our case, although we observed no statistically significant discrepancies, various items were perceived as less clear by the group of participant-judges (older adults).

One critical question we confronted in including older adults as participant-judges was how to create a sample with a consistent level of reading comprehension. To meet this challenge, we adopted stricter protocols than the literature recommended. We avoided possible method bias by using a literacy proxy [60] submitted to a screener, which functioned as a standard criterion [59]. According to this methodological approach, the use of reading comprehension measures for participant-judges should be considered a standard tool in constructing scales [60].

No consensus exists concerning the number of judges required for performing content validation. Nonetheless, a number ranging from five to ten has generally been considered sufficient [34,72]. For this study, the number of participant- and expert-judges was within or greater than the range recommended given the heterogeneity of the sample. The high degree of consensus on the part of both types of judges concerning the clarity of the items and the fact that all construct dimensions were recognized as relevant by the expert-judges confirmed the content validity of our scale. In addition, establishing a system that allowed both types of judges to offer qualitative suggestions on the aspects studied (clarity and relevance), improved the content validation process by permitting the modification of items. In particular, the qualitative feedback stage carried out by some participant-judges provided unique and important contributions for the development of the scale.

We must recognize some limitations in this work. First, during the process of defining the construct dimensions and identifying items, we did not combine deductive and inductive methods. Although the literature review provided the theoretical basis for defining dimensions, the use of qualitative techniques with older adults (representative of our core audience) to define the construct dimensions could have shifted them from the realm of abstraction to the identification of their actual form, reflecting the concrete reality of older adults seeking care. The second limitation pertains to the fact that, as a result of the complexity of the model selected, older adults were not included as participant-judges in the quantitative assessment of relevance. Third, it is reasonable to think that there are individual and contextual factors of the judge-participants (older adults) that may produce variability in the results, and therefore not be replicable. However, the possible individual variability on the perceived clarity of the items was taken into account by maximizing the natural distribution of the judge-participants in the sociodemographic variables, with respect to their population. These sociodemographic variables of the study were age, sex, marital status, level of education, occupation, health needs, and care seeking (see Appendix A). In a general overview, however, we believe there are reasons to presume an acceptable level of consistency in the results related to item clarity for at least four reasons: (a) the sample of judge-participants came from a population for whom it is assumed that they share the same level of experiences in accessing OPPCSs, in terms of their health care needs, care-seeking responses to these services, material conditions and/or organization, etc. Therefore, we can accept that the judge-participants have been exposed to the same characteristics of OPPCSs; (b) the substantive content of the items, due to their theoretical linkage with the literature reviewed, determines that these are consistent indicators of the constructs represented in the sampled population but also in other populations; (c) the possible variability due to reading level was controlled by evaluation and filtering by means of the literacy proxy (see Stage 3. Quantitative evaluation of the content of the scale), and indeed that factor could play a reduced role; and (d) the possible variability related to the construction of the items could also be attenuated by the theoretical and professional adequacy of the expert-judges, described in Section 2.3.1. Participants. Finally, during the fieldwork stage, a high percentage of participant-judges declined to participate because of the fear of COVID-19, as well as concerns about the lack of public security prevailing in the region studied. We recommend that future research be conducted in various population contexts in order to obtain a more universal assessment of the scale, and in times of greater social stability.

## 5. Conclusions

The content-validated instrument Access of Older Adults to Outpatient Primary-Care Health Services Scale (AOAOPHSS) is adequate and useful for comprehensively measuring access to OPPCSs by older adults in Mexico from the perspective of older adults themselves. However, because we are reporting only content validation, the present scale must be considered solely a first step towards a more extensive examination of all its psychometric properties. We recommend expanding the application of the scale to a larger and more geographically diverse group of older adults in order to evaluate its quantitative psychometric properties in general, as well as the validity and dependability of individual items.

## Figures and Tables

**Figure 1 ijerph-19-10102-f001:**
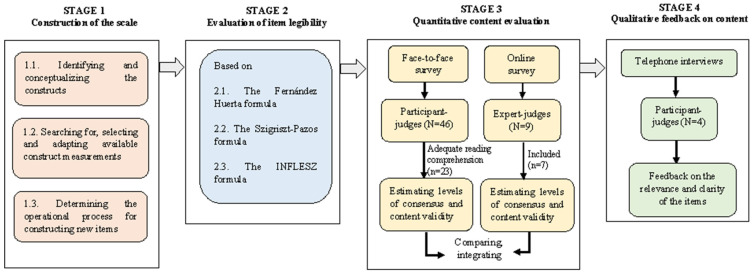
General diagram of the study: processes for the development and evaluation of the scale.

**Figure 2 ijerph-19-10102-f002:**
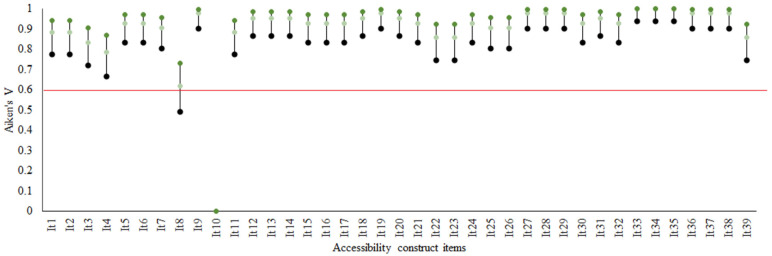
Aiken’s V and confidence intervals for relevance as evaluated by expert-judges (*n* = 7) according to the Accessibility construct.

**Figure 3 ijerph-19-10102-f003:**
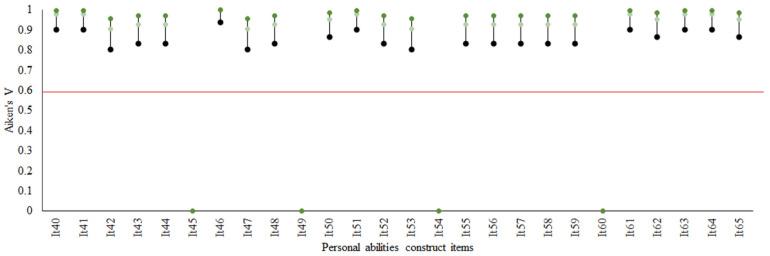
Aiken’s V and confidence intervals for relevance as evaluated by expert-judges (*n* = 7) according to the Personal abilities construct.

**Table 1 ijerph-19-10102-t001:** List of items included in the scale according to the five dimensions of the accessibility construct (*n* = 39).

Construct	Dimension	Operational Definition	Item	Constructed or Adapted From	Element
Accessibility	Approachability	The extent to which outpatient health services disseminate information about their presence, care services, and outreach activities [26,40].	IT1	Constructed	Transparency
			IT2	Constructed	Transparency
			IT3	Constructed	Transparency
			IT4	Constructed	Transparency
			IT5	Constructed	Transparency
			IT6	Constructed	Information
			IT7	Constructed	Outreach
			IT8	Constructed	Outreach
	Acceptability	The extent to which outpatient health services are equitably organized as regards the gender, language, and values of their providers [26].	IT9	ES [43,44]	Language of provider
			IT10	ES [43,44]	Language of provider
			IT11	ES [43,44]	Sex of provider
			IT12	NS [38] ES [42,43]	Values of provider
			IT13	ES [42,43]	Values of provider
	Availability/ accommodation	The extent to which outpatient health services are physically present and have accessible facilities as well as sufficient providers and modes of provision of services [26,40].	IT14	NS [38]	Physically present
			IT15	SM [40] NS [38] ES [45]	Physically present
			IT16	NS [38]	Facilities
			IT17	Constructed	Facilities
			IT18	Constructed	Facilities
			IT19	Constructed	Facilities
			IT20	Constructed	Facilities
			IT21	Constructed	Sufficient providers
			IT22	SM [40] NS [38] ES [42,46]	Modes of provision of services
			IT23	SM [40] NS [38] ES [42,46]	Modes of provision of services
			IT24	SM [40] NS [38] ES [42,46]	Modes of provision of services
			IT25	SM [40]	Modes of provision of services
			IT26	SM [40]	Modes of provision of services
	Affordability	The extent to which outpatient health services allow individuals to mobilize direct, indirect, and opportunity costs as required to obtain care [26,36].	IT27	NS [38,39] ES [48] SM [36]	Direct costs
			IT28	NS [38] ES [47] SM [36]	Direct costs
			IT29	NS [38] SM [36]	Direct costs
			IT30	NS [38]	Indirect costs
			IT31	SM [36]	Indirect costs
			IT32	SM [36]	Opportunity costs
	Appropriateness	The extent to which outpatient health services offer quality, coordinated and continuous care [26].	IT33	NS [38] ES [49]	Quality
			IT34	NS [38] ES [49]	Quality
			IT35	NS [38]	Quality
			IT36	NS [38] ES [46,49]	Continuous care
			IT37	NS [38] ES [46,49]	Continuous care
			IT38	NS [38] ES [49]	Continuous care
			IT39	NS [38] ES [46,49]	Coordination

Note: NS: National survey on health and aging conducted in Mexico [38,39]; SM: Standardized measure for a dimension of accessibility or personal abilities [36,40,41]; ES: Empirical study on access to health services [35,42,43,44,45,46,47,48,49]; IT# = Item number.

**Table 2 ijerph-19-10102-t002:** List of items included in the scale according to the five dimensions of the personal abilities construct (*n* = 26).

Construct	Dimension	Operational Definition	Item	Constructed or Adapted from	Element
Personal abilities	Ability to perceive	The capacity of older adults to identify the need for health according to their knowledge and beliefs regarding the health–disease cycle [26,35].	IT40	SM [41]	Beliefs
			IT41	NS [38]	Beliefs
			IT42	NS [38]	Beliefs
			IT43	SM [41]	Beliefs
			IT44	SM [41]	Knowledge of health
			IT45	Constructed	Knowledge of health
	Ability to seek	The capacity of older adults to recognize available health-care options in accordance with their values and personal autonomy for choosing [26,35].	IT46	ES [35]	Values
			IT47	ES [35]	Autonomy
			IT48	ES [35] SM [41]	Autonomy
			IT49	Constructed	Autonomy
	Ability to reach	The capacity of older adults to overcome occupational, personal mobility and/or transportation barriers as regards travel to a health-care facility [26,35].	IT50	NS [38] ES [35]	Notion of mobility
			IT51	Constructed	Notion of mobility
			IT52	NS [38]	Available transportation
			IT53	Constructed	Available transportation
			IT54	Constructed	Occupational flexibility
	Ability to pay	The capacity of older adults to meet the direct and indirect costs of care without negatively impacting their basic needs [26,35].	IT55	NS [39] ES [49]	Income
			IT56	NS [39] ES [49]	Income
			IT57	NS [39]	Income
			IT58	NS [39] ES [49]	Savings
			IT59	NS [39] ES [49]	Savings
			IT60	NS [39]	Loans
	Ability to engage	The capacity of older adults to actively participate in decision-making and adhere to treatment [26,35].	IT61	Constructed	Adherence
			IT62	Constructed	Adherence
			IT63	Constructed	Adherence
			IT64	Constructed	Empowered
			IT65	ES [35]	Empowered

Note: NS: National survey on health and aging conducted in Mexico [38,39]; SM: Standardized measure for a dimension of accessibility or personal abilities [36,40,41]; ES: Empirical study on access to health services [35,42,43,44,45,46,47,48,49]; IT# = Item number.

**Table 3 ijerph-19-10102-t003:** Aiken’s V and confidence intervals for clarity as evaluated by participant-judges (*n* = 23) and expert-judges (*n* = 7).

Clarity													
Construct Accessibility	Participant-Judges			Expert-Judges			Construct Personal Abilities	Participant-Judges			Expert-Judges		
	Aiken’s V	Lower	Upper	Aiken’s V	Lower	Upper		Aiken’s V	Lower	Upper	Aiken’s V	Lower	Upper
*Approachability*							*Ability to perceive*						
It1	0.695	0.611	0.767	0.785	0.665	0.870	It40	0.708	0.624	0.779	0.857	0.746	0.924
It2	0.708	0.624	0.779	0.738	0.615	0.833	It41	0.805	0.729	0.864	0.882	0.776	0.941
It3	0.695	0.611	0.767	0.667	0.540	0.773	It42	0.805	0.729	0.864	0.833	0.719	0.907
It4	0.708	0.624	0.779	0.667	0.540	0.773	It43	0.793	0.715	0.853	0.833	0.719	0.907
It5	0.728	0.645	0.797	0.833	0.719	0.907	It44	0.773	0.693	0.836	0.762	0.640	0.852
It6	0.718	0.635	0.788	0.810	0.693	0.890	It45	--	--	--	--	--	--
It7	0.685	0.601	0.758	0.857	0.746	0.924							
It8	0.695	0.611	0.767	0.882	0.776	0.941							
*Acceptability*							*Ability to seek*						
It9	0.643	0.557	0.720	0.785	0.665	0.870	It46	0.837	0.765	0.891	0.928	0.834	0.971
It10	---	---	---	---	---	---	It47	0.740	0.659	0.808	0.857	0.746	0.924
It11	0.620	0.534	0.699	0.738	0.615	0.833	It48	0.750	0.669	0.816	0.833	0.719	0.907
It12	0.708	0.624	0.779	0.762	0.640	0.852	It49	---	---	---	---	---	---
It13	0.760	0.680	0.825	0.810	0.693	0.890							
*Availability/accommodation*							*Ability to reach*						
It14	0.750	0.669	0.816	0.762	0.640	0.852	It50	0.793	0.715	0.853	0.857	0.746	0.924
It15	0.783	0.704	0.845	0.738	0.615	0.833	It51	0.815	0.740	0.872	0.833	0.719	0.907
It16	0.760	0.680	0.825	0.738	0.615	0.833	It52	0.773	0.693	0.836	0.810	0.693	0.890
It17	0.783	0.704	0.845	0.762	0.640	0.852	It53	0.773	0.693	0.836	0.857	0.746	0.924
It18	0.773	0.693	0.836	0.833	0.719	0.907	It54	---	---	---	---	---	---
It19	0.750	0.669	0.816	0.952	0.865	0.984							
It20	0.728	0.645	0.797	0.928	0.834	0.971							
It21	0.760	0.680	0.825	0.810	0.693	0.890							
It22	0.728	0.645	0.797	0.928	0.834	0.971							
It23	0.695	0.611	0.767	0.952	0.865	0.984							
It24	0.728	0.645	0.797	0.905	0.804	0.957							
It25	0.740	0.659	0.808	0.952	0.865	0.984							
It26	0.718	0.635	0.788	0.882	0.776	0.941							
*Affordability*							*Ability to pay*						
It27	0.783	0.704	0.845	0.905	0.804	0.957	It55	0.773	0.693	0.836	0.905	0.804	0.957
It28	0.773	0.693	0.836	0.857	0.746	0.924	It56	0.740	0.659	0.808	0.810	0.693	0.890
It29	0.773	0.693	0.836	0.905	0.804	0.957	It57	0.708	0.624	0.779	0.882	0.776	0.941
It30	0.773	0.693	0.836	0.857	0.746	0.924	It58	0.708	0.624	0.779	0.905	0.804	0.957
It31	0.783	0.704	0.845	0.952	0.865	0.984	It59	0.750	0.669	0.816	0.905	0.804	0.957
It32	0.773	0.693	0.836	0.833	0.719	0.907	It60	---	---	---	---	---	---
*Appropriateness*							*Ability to engage*						
It33	0.783	0.704	0.845	0.928	0.834	0.971	It61	0.815	0.740	0.872	0.833	0.719	0.907
It34	0.773	0.693	0.836	0.952	0.865	0.984	It62	0.805	0.729	0.864	0.857	0.746	0.924
It35	0.760	0.680	0.825	0.952	0.865	0.984	It63	0.805	0.729	0.864	0.905	0.804	0.957
It36	0.773	0.693	0.836	0.810	0.693	0.890	It64	0.773	0.693	0.836	0.952	0.865	0.984
It37	0.805	0.729	0.864	0.810	0.693	0.890	It65	0.793	0.715	0.853	0.785	0.665	0.870
It38	0.825	0.751	0.881	0.833	0.719	0.907							
It39	0.825	0.751	0.881	0.857	0.746	0.924							

Note: The underlined confidence interval limits indicate that Aiken’s minimum criterion was not met.

## Data Availability

Not applicable.

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
