# Peer review of "Content Validity of a Scale Designed to Measure the Access of Older Adults to Outpatient Health Services"

_ijerph, 2022, doi:10.3390/ijerph191610102_

Round 1

Reviewer 1 Report

Congratulations to the authors for the effort and quality of their work.

1. Introduction

The background is clear and exhaustive. The rationale is coherent and relevant to fill the knowledge gap in the area.

2. Materials and Methods

Very clear and sequential methodological structure. I only suggest in figure 1 to order the points contained in stage 1 as: 1.1, 1.2 and 1.3 from top to bottom.

3.- Results

Very clear presentation.

4. Discussion

Very interesting arguments. Contains all the necessary elements.

5. Conclusions

Clear and direct.

Author Response

Response:

Thank you for considering our manuscript as a significant contribution to the field.

 Change:

In Figure 1, the points were ordered in stage 1 (see line 100, page 3 of the manuscript).

Reviewer 2 Report

The manuscript is interesting and cover a important topic, however, I have some question about the work

1.- I understood that the scale was constructed and evaluated in a small population, it could have the same significance in a big population.

2.- Is possible to know the complete instrument constructed and validated? Is possible to present it in a supplementary material.

3.- The instrument constructed could give different results depending on the people education, where people live and other factors. where these factors taking into account.

Author Response

Responses

We are very grateful for your comments. They help us improve our study. Below you can find the answers to each question.

Response 1:

We agree with the comment. This aspect is included in the last paragraph of the conclusion.

Response 2:

A supplementary section on the instrument has been added (see line 530, page 14 of the manuscript).

Response 3:

The possible individual variability on the perceived clarity of the items was taken into account by maximizing the natural distribution of the participants in the sociodemographic variables, with respect to their population. These sociodemographic variables of the study were age, sex, marital status, level of education, occupation, health needs and care seeking. To describe the distribution of the participants across these variables we created a descriptive sociodemographic table in the new supplementary material (Table S1, see line 533, page 14 of the manuscript article).

In a general overview, however, we believe there are reasons to presume an acceptable level of consistency in the results related to item clarity. Specifically: a) the sample of judge-participants came from a population for whom it is assumed that they experience the same level of experiences with first-level outpatient services. This is in terms of frequency according to health condition, care responses of these services, material conditions and/or organization. Therefore, we can accept that the judge-participants have been exposed to the same characteristics of access to first-level ambulatory health services; b) the content of the items, due to their theoretical linkage with the literature reviewed, may be consistent indicators of the constructs represented in the sampled population but also in other populations; c) the possible variability due to the level of reading comprehension was controlled by evaluation and filtered by means of the literacy proxy (see Stage 3. Quantitative evaluation of the content of the scale), and indeed that factor could have played a reduced role; and d) the possible variability related to item construction could also be attenuated by the theoretical and professional adequacy of the expert-judges, described in section 2.3.1 Participants.

Change:

In agreement with response 3, a limitation statement was added from line 494 to line 515 on page 13 and 14 of the manuscript.  

Reviewer 3 Report

Dear authors!

Thank you for your research. I have few questions and hope your answers will clarify it.

1. Why didn’t you make a null hypothesis?

2. Were surveys you've used valid and if yes please write about it in the text. Also if not explained how did you verified it.

3. Table 3 looks too big, try to remake it of may be add a graphic as a comment.

Author Response

Responses

We are very grateful for your comments. They help us improve our study. Below you can find the answers to each question.

Response 1:

Given that the context of the study is fully exploratory, in terms of developing items of a new measure, we can dispense with a null hypothesis. However, we hope to be able to generate hypotheses in the next manuscript about the internal structure of the questionnaire. 

Response 2:

Following the recommendation, a reference was added in the text (see line 148 on page 4 of the manuscript).

Response 3:

Following the recommendation, Table 3 was replaced by two figures that improve the presentation of the results.

Change:

Figures 2 and 3 were added in replacement of Table 3 (see line 380, page 11 of the manuscript).